# Maternal Malnutrition in the Etiopathogenesis of Psychiatric Diseases: Role of Polyunsaturated Fatty Acids

**DOI:** 10.3390/brainsci6030024

**Published:** 2016-07-27

**Authors:** Maria Grazia Morgese, Luigia Trabace

**Affiliations:** Department of Clinical and Experimental Medicine, University of Foggia, Foggia 71122, Italy; luigia.trabace@unifg.it

**Keywords:** polyunsaturated fatty acids, depression, anxiety, autism spectrum disorder, schizophrenia spectrum disorder, attention deficit disorder, maternal malnutrition

## Abstract

Evidence from human studies indicates that maternal metabolic state and malnutrition dramatically influence the risk for developing psychiatric complications in later adulthood. In this regard, the central role of polyunsaturated fatty acids (PUFAs), and particularly *n*-3 PUFAs, is emerging considering that epidemiological evidences have established a negative correlation between *n*-3 PUFA consumption and development of mood disorders. These findings were supported by clinical studies indicating that low content of *n*-3 PUFAs in diet is linked to an increased susceptibility to psychiatric disorders. PUFAs regulate membrane fluidity and exert their central action by modulating synaptogenesis and neurotrophic factor expression, neurogenesis, and neurotransmission. Moreover, they are precursors of molecules implicated in modulating immune and inflammatory processes in the brain. Importantly, their tissue concentrations are closely related to diet intake, especially to maternal consumption during embryonal life, considering that their synthesis from essential precursors has been shown to be inefficient in mammals. The scope of this review is to highlight the possible mechanisms of PUFA functions in the brain during pre- and post-natal period and to evaluate their role in the pathogenesis of psychiatric diseases.

## 1. Introduction

Maternal malnutrition has been linked to the development of different neuropsychiatric disorders in infancy (Autism Spectrum Disorder (ASD)) [1], in childhood (Attention deficit disorder (ADHD)) [2] ref, in adolescence (Schizophrenia spectrum disorders (SSD)) [3,4] or in adulthood (depression and anxiety) [5]. In searching for dietary factors that may be linked to the development of these neuropsychiatric disorders, the role of polyunsaturated fatty acids (PUFAs) is becoming more evident considering that the analysis of lipid composition, the-so called lipidomics, is gaining great attention in psychiatric illness field [6].

PUFAs have been shown to be crucially involved in normal brain development and functioning [7]. These molecules can be classified based on the number of carbon atoms separating the first double bond of the carbon chain from the terminal methyl end (namely omega): *n*-3, *n*-6, and *n*-9. PUFAs with a relevant biological function belong to the *n*-3 or *n*-6 family. The most relevant molecules are docosahexaenoic acid (DHA, 22:6*n*-3), *n*-3 family, and arachidonic acid (AA, 20:4*n*-6), for *n*-6 family, representing 40% and 50% of brain PUFAs, respectively [8,9,10,11]. Another important lipid of *n*-3 family is eicosapentaenoic acid (EPA, 20:5*n*-3) [12,13,14,15], whose brain concentration is significantly lower [16,17] considering that its β-oxidation reaction occurs very rapidly [18]. Isomers of docosapentaenoic acid (DPA, 22:5), either *n*-6 or *n*-3, also appear relevant considering that have been shown to partially substitute DHA in condition of scarce *n*-3 PUFA intake [19,20,21]. It has been described that the ratio *n*-6/*n*-3 has dramatically increased during recent decades because of profound changes in lifestyle and changes in food sources. Such ratio, that during early 20th century was estimated around a value of 1, is now ranging in Western countries around 15 or even more [22,23,24]. AA precursor, linoleic acid (LA 18:2*n*-6), is found in soybean, canola, corn, safflower, sunflower and cottonseed, while AA is present principally in meat or products derived from animal fed with high *n*-6 content diets [22,25]. On the other hand α-linolenic acid (ALA 18:3*n*-3), the precursor of DHA and EPA, is highly concentrated in walnut, rapeseed, flax seeds and chia [26]. DHA and EPA nutritional source is mostly seafood [27]. Main sources of these molecules are summarized in Table 1. Thus, changes in industrial food preparations may be responsible of the abnormal change in *n*-6/*n*-3 PUFA ratio. Importantly, the high ratio *n*-6/*n*-3 is particularly unfavorable for proper central nervous system (CNS) functioning. LA and ALA compete with each other for enzymatic convention and, although *n*-3 in healthy conditions are favored [28,29], high LA levels can interfere with this enzymatic conversion reducing DHA levels [30]. Importantly, maternal fatty acid composition during gestation and lactation influences not only fetal and infant blood levels, but also lipid compositions in maturing tissues [31]. In the present work, we aim at summarizing the consequences of maternal low PUFA intake on the developing of neuropsychiatric disorders.

### Biological Role of PUFAs in Brain Development

Mammalian brain is very rich in lipids (60%/65%) comprising saturated, mono- and polyunsaturated fatty acid. DHA is the major *n*-3 PUFAs in mammalian brain and particularly in gray matter. It has been shown that DHA accounts for around 10%–20% of total fatty acid composition of prefrontal cortex (PFC) in adulthood [32,33].

DHA and AA source can be derived directly from diet or through metabolic conversion of their essential precursors ALA and LA, respectively. The enzymatic conversion relies on the action of Δ5 and Δ6 desaturases and on the elongase, enzymes responsible of converting LA into AA e ALA into EPA and DHA. Although in mammals these enzymatic reactions are possibly occurring, in humans this conversion is quite inefficient, thus tissue concentrations of these lipids reflect diet intake [34]. Confirming such an hypothesis, it has been shown that DHA concentrations in erythrocytes [35], breast milk [36,37], and cortical gray matter [38,39] are noticeably increased consequently to preformed DHA consumption. Furthermore, it should be underlined that in rodents and other animal species the conversion rate of DHA from ALA has been shown to be more efficient in contrast to humans [40], thus this aspect should be considered when trying to compare clinical and preclinical observations.

In humans, during the prenatal period DHA accumulates rapidly particularly during the third trimester of gestation, when synaptogenesis, differentiation, neurogenesis and expansion of gray matter occur [41,42]. At early post-natal life before weaning, human brain greatly accumulates DHA. It has been estimated that DHA is very selectively up-taken through the blood–brain barrier during this period, considering that its cerebral increase represents 50% of total body amount [7,43,44].

Childhood and adolescence are critical periods for brain maturation, and particularly for the creation of connections between brain areas involved in mediating attention and executive functions, such as frontal lobe regions, and brain areas implicated in the regulation of emotions and mood, such as limbic areas [45,46,47]. Significantly, DHA accrual at PFC level increases sharply with circuit maturation and, in preterm delivery, DHA deficiency has been associated with permanent impairment of this neural network linked to development of psychiatric disorders later in life [32]. In rodents, it has been reported that DHA levels increase up to five fold from early until late gestation [48]. *n*-3 PUFAs have been reported to exert their central action by modulating synaptogenesis and neurotrophic factor expression [49]. Indeed, these lipids are essential to assure proper brain functioning considering that DHA content is higher in cerebral rather than any other tissue [50]. Furthermore, these lipids regulate membrane fluidity particularly modulating phospholipids composition, such as phosphathidyl serine concentrations [51]. This phospholipid regulates many important enzymes, such as protein kinase C, which controls the function of structural proteins involved in neurite outgrowth and neurotransmitter release [32]. Figure 1 graphically represents main PUFA targets in neurodevelopment (Figure 1). Furthermore, deficiency of DHA during the perinatal period leads to increased cytokine production by altering microglia phenotype [52], thus resulting in neuroinflammatory profile. On the other hand, the role played by oxidative and nitrosative stress in brain development and in early life alterations that may lead to psychiatric disorders is emerging [53]. In this respect, it has been shown that a transient accumulation of oxidative products occurs in neonatal rat brain [54]. Furthermore, after birth brain oxygen concentration sharply increases compared to fetal environment, thus creating a condition of oxidative stress. In addition, after birth expression of both neuronal and inducible nitric oxide synthases reaches a peak on the fifth postnatal day [55] and products of nitrosative stress, such as nitrotyrosine, have been detected during early post-natal period [55,56]. Early postnatal period is characterized by apoptotic elimination of both neurons and glia over abundantly proliferated [57] and then apoptotic cells are phagocyted by microglia resulting in oxidative and nitrosative species production [58]. Furthermore, in humans, brain development has been shown to involve basal levels of oxidative stress [59]. Interestingly, it has been demonstrated that consumption of diet rich in trans fats at a pre- and perinatal period in rats, corresponding to poor *n*-3 availability, leads to higher lipoperoxidation levels and lower cell viability in cortex, along with reduced catalase activity in striatum and hippocampus, and increased generation of reactive species in striatum [60]. In addition, *n*-3 PUFAs may influence central redox homeostasis by increasing the levels of neuroprotectin D1, a protectin produced in neurological tissues and derived by DHA enzymatic conversion. Neuroprotectin D1 can in turn regulate pro-survival and repair activity through Bcl-2-related molecule activation and BAD, BAX, and Bid inhibition [61]. Interestingly, it has been reported that placental ROS may tonically suppress fetal growth [62]. In this regard, maternal *n*-3 PUFA dietary supplementation was shown to reduce placental levels of F2-isoprostanes, a highly reliable marker of oxidative damage associated with enhanced fetal and placental growth [63].

The role *n*-6 PUFAs in brain maturation and development is much less studied. However, the main *n*-6 derivative, AA, plays crucial physiological functions considering that it is an important precursor of bioactive mediators and it has been shown to activate protein kinases and ion channels leading to enhanced synaptic transmission [5]. Importantly, AA-derivatives, such as endocannabinoids, have been shown to act as retrograde messengers in hippocampal long-term potentiation [6,7] and to be involved in cortical neuron migration [8]. Furthermore, endocannabinoids regulate specific molecular events related to neural development considering that receptors and ligands of the endocannabinoid system critically appear during brain development in atypical locations [64]. Endocannabinoids have also been shown to regulate neuronal migration and to induce migration of GABA-expressing interneurons in the embryonic cortex. Furthermore, endocannabinoids, along with their related lipid mediators, have been implicated in the regulation of commitment of neural progenitor survival [65,66] and in the development of synaptic connectivity in brain maturation [67,68].

On the other hand, a high ratio *n*-6/*n*-3 is particularly unfavorable for proper central nervous system (CNS) functioning. Indeed, when diets are particularly poor in *n*-3 PUFAs, endoreticulum, as well as peroxisomes, metabolically produce *n*-6 derivatives, namely *n*-6 docosapentaenoic acid (22:5*n*-6), whose insertion into biological membranes, in place of DHA, is considered the main process responsible of CNS dysfunction [11]. Interestingly, it has been reported that DHA accrual during early childhood is strictly dependent on *n*-6 PUFA consumption. Indeed, when DHA or in general *n*-3 PUFA intake is limited in presence of a diet rich in LA, infant brain accumulates *n*-6 PUFAs and less *n*-3 PUFAs [69]. This event could lead to significant implications considering that *n*-6 PUFAs, and then AA, are precursors of important molecules, the eicosanoids, involved in inflammation and immune system regulation. In this regard, high *n*-6 PUFA consumption during early development and during infancy has been associated with an increased risk of allergic syndromes indicating that these lipids may play a role in programming of immune system [70].

The effect of maternal PUFA intake in fetal programming is becoming evident, although the exact mechanisms still need to be fully elucidated. Many studies, either epidemiological or preclinical, are now indicating the deleterious effects that low maternal consumption of *n*-3 PUFA may have in offspring, particularly in relation to psychiatric disorders.

## 2. Early Life Programming: Models and Mechanisms

A large body of recent literature has focused the attention on the study of how the exposure to environmental factors during pre-, peri- and post-natal period can influence health status and disease occurrence later in life. The effects of this early exposure can persist later in life, thus this phenomenon has been called “early life programming” (ELP). The fetal origins of adult disease model has its roots on Barker’s observations that associated undernutrition of the fetus, and thus low body weight at birth, with an increased risk of cardiovascular disease, diabetes and metabolic syndrome later in life [71]. ELP has been extensively studied in humans, however epidemiological studies are often affected by many biases and in this regard animal models may result very helpful.

In animal models, ELP has been studied by using several paradigms, such as maternal protein restriction, maternal caloric restriction, maternal iron restriction, maternal glucocorticoid exposure, maternal hypoxia, uterine placental ligation and many others [72]. All these models lead in offspring to low body weight at birth or preterm delivery and have been associated with several pathological conditions later in life.

Studies on animal models based upon maternal dietary restriction, such as protein [73], caloric or iron restrictions, have found profound changes in offspring leading, among other diseases, to type 2 diabetes and glucose tolerance [74], hypertension [75,76,77], and hypothalamic-pituitary-adrenal (HPA) disturbances [78]. Furthermore, alterations in the energy balance, in placental cytokine [79], in renal morphology [80], impairment of plasticity and maturation of the brain [81], and in pancreatic functions [82] have also been reported. Although many models have been established by reducing nutrient intake in maternal diet, it has been shown that also high caloric diets, and particularly high fat diets, are effective in inducing long-lasting effects in offspring. In particular, excessive nutrition during gestation and lactation has been associated with many diseases, such as metabolic syndrome [83], pro-atherogenic lesions [84], hyperleptinemia [85], anxiety-like behavior and perturbation in serotonergic system [86] as well as impairment in learning and memory function by interfering with hippocampal gene expression [87].

Maternal glucocorticoid exposure is considered a model of prenatal stress [88]. Chronic stress during pregnancy has been associated with increased risk of development of mental illness later in life [89], as well as maternal dietary factors. High glucocorticoids during pregnancy have been associated with higher incidence of schizophrenia, increased left or mixed handedness, reduction in cerebral asymmetry and anomalies in brain morphology [90]. Interestingly, stress response in adulthood depends on perinatal programming. After in utero exposure to high glucocorticoids or after prenatal stress, decreased glucocorticoid receptors (GR) at hippocampal level have been found. Such alterations were accompanied by high circulating glucocorticoids and adrenocorticotropic hormone (ACTH), and it has also been reported an increased response to stressful stimuli [91,92]. Furthermore, prenatal glucocorticoid exposure leads to increased corticotrophin releasing factor (CRF) expression in the amygdala, a brain area crucially involved in fear conditioning and stress response [92].

In trying to understand mechanisms that predispose to disease development later in life, epigenetic studies provided interesting results. In particular, prenatal stress has been shown to induce hypermethylation in the placenta and hypomethylation in the fetal hypothalamus, respectively, of *Hsd11b2*, a gene encoding for the enzyme 11β-hydroxysteroid dehydrogenase type 2 responsible of converting cortisol into an inactive metabolite and thus providing fetal protection from cortisol [93]. Such genetic modification corresponds to an altered enzymatic activity, thus enhancing stress responses in offspring [94,95,96]. Furthermore, permanent alterations of DNA methylation status following prenatal stress have been reported, such as hypermethylation and hypomethylation, respectively, in *Gr* noncoding exon_17_ and in the *Crf* promoter in hypothalamus [97]. These results appear of relevant interest considering that altered stress response has been reported following maternal poor intake of PUFAs, particularly *n*-3 PUFAs [98,99]. Significantly, alteration in stress response, both in human and animal models, has been linked to many psychiatric diseases, particularly depression [100] and psychosis [101].

Furthermore, supplementation with *n*-3 PUFAs during pregnancy has been reported to modulate global methylation levels and to imbalance the ratio of T helper cell 1 and 2 in infants, thus altering cytokine production. Neuroinflammation has been indicated in this regard as a possible bridge between neuropsychiatric disorders and increased stress response [102]. Therefore, in the following section we will revise the literature focusing the attention to alteration in maternal PUFA intake and neuropsychiatric diseases.

### 2.1. Role of n-3 PUFAs on Depression and Anxiety

Recent evidences have pointed out that the prevalence of depression has reached epidemic proportions in last decades. Such an increase has been linked to many environmental factors; among these, the influence of dietary factors has gained great attention. In particular, it has been reported the *n*-3 PUFA content in diet was inversely correlated to the development of depressive symptoms [103,104,105], and such association was substantial and stronger in women [106]. Clinical studies indicating that low content of *n*-3 PUFAs in diet is linked to an increased susceptibility to psychiatric disorders like depression supported these findings [107,108]. *n*-3 supplementation alone or in adjunctive therapies showed positive results in the treatment of this mental disorder [109]. In a recent updated meta-analysis of randomized controlled trials investigating the efficacy of *n*-3 PUFA treatment in depressive disorders such beneficial effects were further endorsed [110], as also indicated by recent guidelines from British Association from Psychopharmacology [111]. However, contrasting results have also been reported in epidemiological outcomes and significant associations were lost in some cases after normalization of data and adjusting for lifestyle confounders [110,112]. Animal studies have yielded results that are more consistent. Larrieu and Colleagues [113] demonstrated that in mice maternal depauperation of *n*-3 PUFAs, followed by a post-natal diet with the same deficiency, led to increased anxiety- and depressive-like behavior in the offspring. Moreover, such behavioral impairment was associated with ablation of endocannabinoid-mediated long-term synaptic depression in the prelimbic PFC and accumbens, and alterations in cannabinoid receptor signaling [98,114]. In addition, pharmacological blockade of anandamide breakdown, a PUFA metabolite, was shown to reduce fine motor and working memory impairments and anxiety-like behaviors, as well as to slow neurodegeneration and amyloid precursor protein production, and was able to upregulate stress-responsive growth factors and heat shock proteins [115].

On the other hand, deficiency in *n*-3 PUFA intake, especially in perinatal period, is linked to decreased brain derived neurotrophic factor (BDNF) content [52] and low BDNF levels have been described after prenatal stress [116]. Glucocorticoids have been related to such an effect, since corticosterone is able to down-regulate both mRNA and protein BDNF [117,118]. In this light, mice over-expressing glucocorticoids showed an increased anxiety-like behavior [119] and Larrieu and Colleagues have demonstrated that *n*-3 PUFA deficiency can influence neuronal cortical morphology and depressive-like behavior through corticosterone secretion [98]. Furthermore, they showed that a condition of poor *n*-3 diet intake induces a phenotype comparable to one induced by chronic social defeat stress and high corticosterone levels were also described [98]. Moreover, very recently it has been demonstrated that GR functioning is altered only in an area crucially involved in emotional behavior, such as PFC, of mice exposed to poor *n*-3 PUFA diet from conception, further supporting the major role of these lipids in cortical maturation [120].

In this regard, we have recently reported that lifelong nutritional deficiency in *n*-3 PUFAs, from gestation until 8 weeks of life in rats, leads to a consistent depressive-like and anxiety-like profile. Furthermore, these behaviors were associated with a hyper-activation of HPA axis, with increased plasma corticosterone levels and high hypothalamic CRF. In addition, only in PFC reduced serotonin levels were found, thus supporting behavioral with neurochemical outcomes [99]. In further agreement with the depressive-like profile, we also found in our model increased plasma levels of soluble amyloid beta (Aβ). This peptide, usually linked to Alzheimer’s disease, has been shown to be increased in depressed patients [121,122,123,124]. In this regard, we have previously found that Aβ, intracerebroventricularly injected is able to evoke a depressive-like profile in rats associated with reduced cortical serotonin and neurotrophin levels [125]. In addition, in this model, we found alteration in HPA axis and increased cortical noradrenaline and glutamate levels [126,127,128]. Therefore, based on our experience we suggest that *n*-3 PUFA deficiency by increasing Aβ levels and perturbing HPA axis functioning can predispose to alteration in emotional profile. In good agreement, a dramatic increase also in Alzheimer’s disease (AD) prevalence has been documented either in developing or in industrialized eastern countries, such as Japan, after Westernization of national diets, and thus after shifting *n*-3 in favor of high *n*-6 PUFA intake [129]. A bridge between these pathologies is represented by neuroinflammation, considering that central serotonergic system is negatively affected by inflammation. Interestingly, DHA was shown to reduce eicosanoid synthesis along with AA production [130,131], and to decrease inflammatory cytokine levels [132]. Moreover, *n*-3 PUFAs, given in adulthood to rats, were shown to reduce the production of interleukin 1β and interferon γ induced by central injection of Aβ [133], molecules strictly linked to depression [134,135,136]. In this regard, *n*-3 supplementation was shown to be potentially beneficial in preventing and/or treating several chronic inflammatory conditions and neurodegenerative diseases such as AD [137,138]. Moreover, a randomized, double-blind, placebo-controlled study evidenced the positive effect of DHA and EPA adjunctive therapy on depressive symptoms in mild to moderate AD patients [139]. Although *n*-3 supplementation has been shown to improve cognitive and depressive symptoms in AD [140], no data are available about the role of *n*-3 deficiency in maternal diet and later development of degenerative symptoms. Future studies are surely warranted.

### 2.2. Role of Maternal n-3 PUFAs on Schizophrenia Spectrum Disorders

Schizophrenia Spectrum Disorders are now accepted as a neurodevelopmental disorders. Some etiopathogenic origins of these disorders have been found as early as the prenatal period [141]. Early life and prenatal stress have been implicated as potential risk factors for development of later in life psychotic events both in human and in animal studies [89,142]. Gestational DHA deficiency has been shown to potentially contribute to the etiopathogenesis of SSD. Indeed, prenatal DHA deficiency has been linked to alteration in neurotransmission, such as alteration in dopaminergic system functioning, and neurocognitive impairments [143,144,145,146], with deficits in cortical maturation and attention [147,148,149,150,151,152]. These events have also been reported in schizophrenic patients [153]. In addition, *postmortem* studies in schizophrenic patients indicated modification in the expression of genes responsible for regulating the biosynthesis of fatty acids at PFC levels [154]. Moreover, chronic antipsychotic treatments can also upregulate PUFA levels in red blood cell of patients with psychosis [155,156]. Lowered rate of conversion of EPA into DHA has also been observed in schizophrenia [157] along with high catabolism of *n*-3 PUFAs [155,158,159]. It was reported that disturbances in PUFA metabolism occurs very early in antipsychotic naive patients [160,161].

Increased offspring neurocognitive functioning has been demonstrated in many double-blind, placebo-controlled randomized clinical trials conducted in mothers receiving either supplementation with DHA or consuming DHA-rich foods [147,148,149,162]. Furthermore, reduced consumption of food rich in *n*-3 PUFAs was associated with lower verbal IQ, diminished prosocial behavior, suboptimal fine motor ability, and impaired social and verbal development [163].

Interestingly, it has been shown that perturbations in dopaminergic system, a neurotransmitter system strictly involved in SSD, can be completely reversed by adequate levels of *n*-6 and *n*-3 PUFAs consumption only at lactation stage, while at post-weaning period the reversal was only partial [146]. Thus, these results indicate that the correct amount of PUFA intake needs to be early assured in the diet of breastfeeding mothers, as well as in infant milk formulas. Abnormalities of the mesocortical dopamine system have been proposed to contribute to the pathophysiology of schizophrenia along with dysfunction of PFC [164,165]. In a rat model, prenatal *n*-3 PUFA deficiency was shown to impair the mesocortical and mesolimbic dopaminergic systems [43,166,167] and further *n*-3 supplementation was not able to completely reverse such alterations associated with fronto-limbic dysfunction in PFC [166,167,168,169]. Again, these partial reversal effects could reflect brain-region specificity of *n*-3 supplementation efficacy. In good agreement, restoration of *n*-3 DHA levels through supplementation varies in the PFC compared to the striatum and cerebellum [170]. The specificity of this brain region in PUFA turnover may also be reflected in behavioral expression of PUFA deficiency. Frontal region related functions, particularly impaired in patients affected by SSD, such as executive dysfunction may be more vulnerable to PUFA ratio changes and only show partial response to *n*-3 supplementation [150,152,171].

### 2.3. Role of n-3 PUFAs on Autistic Spectrum Disorders

Autistic Spectrum Disorders are characterized by defects in communication and social behavior and are associated with repetitive behaviors [172]. The etiopathogenesis of ASD is still unknown, although events occurring at pre- and perinatal timeframe appear crucial [173,174]. Thus, genetic predisposing factors associated with environmental risk factors seem very likely interfering [175,176]. In this regard, a maternal specific polymorphism of serotonin transporter that may affect the risk for ASD if associated with exposure to prenatal stress [177] and different types of prenatal stressors emerged as possible potential contributor to autism [178,179,180]. Furthermore, it has been reported that the timing of stressor exposure, embryonal age and neuroanatomical findings in cerebellum well correlate [181]. In addition, research has pointed out that several maternal prenatal factors may be implicated [182], and in particular maternal nutritional status has gaining attention. However, in regard to PUFA intake during pregnancy, very few data are available. It has been shown that very low maternal intake of *n*-3 PUFAs was associated with an increased risk of ASD development [172]. In the same study, Authors also found a protective effect of *n*-6 PUFAs, usually associated with increased inflammatory response. Thus, as the same Authors stated, possible confounding factors may have affected the results and more accurate studies need to be performed. Indeed, in rodents, it has been reported that lifelong nutritional exposure to a diet rich in *n*-6 PUFAs associated with prenatal stress contributes to the production of autistic-like behavior [183]. Many links associate PUFAs with ASD. In particular, impairments in synapse homeostasis are considered a novel field of research [184] and *n*-3 PUFA deficiency has been implicated in synapse plasticity [32]. In addition, alterations in myelination and aberrant long-range brain connectivity have also been proposed as pathological hallmarks of ASD [185,186,187,188,189]. Interestingly, increased *n*-6/*n*-3 PUFA ratio was found leading to profound changes in myelination and thus in connectivity [190,191]. ASD has an incidence 3–4 times higher in males than in females [192] and PUFA metabolism studies revealed that conversion of ALA into DHA is negligible in males while it occurs in females at rate of 9% [193,194], further indicating a potential role for DHA in this disorder. From a neurochemical point of view, alterations in GABAergic transmission have been postulated in ASD [195,196,197] and modulation of GABAegic receptor functions has been described for DHA [198,199,200]. Furthermore, alterations in neurotrophins, and in particular BDNF, and alterations in PUFA metabolism have also been reported in ASD [201]. Deficiency in *n*-3 PUFA intake, especially in perinatal period, is linked to decreased BDNF content [52], thus more studies focused on the role of maternal low PUFA consumption and the risk of ASD development are surely warranted.

### 2.4. Role of n-3 PUFAs on Attention Deficit Disorder

Attention Deficit Disorder is mainly occurring in children and is characterized by inattentive, impulsive and hyperactive behavior. The link between ADHD disorders and PUFA consumption originates from the study of Stevens et al. (1995) that found significant reduction in plasma DHA and EPA levels in affected patients [202]. Furthermore, some symptoms that have been found in conditions of essential fatty acid deficiency are commonly shared with ADHD. In particular, the so called thirst/skin symptoms characterized by excess thirst, frequent urination, dry skin, dry hair, dandruff, and brittle nails, have also been reported in ADHD subjects [202,203]. Moreover, it has been proposed that maternal low consumption of *n*-3 PUFAs or low dietary intake after birth may cause ADHD [204,205]. Unfortunately, human studies have led to inconclusive results indicating either no improvement after 16 weeks period DHA (345 mg/die) supplementation [206] or beneficial effect following 12 weeks of supplementation with a combination of EPA (186 mg/die) and DHA (480 mg/die) [207]. In 2012, a systematic review published on the Cochrane Systematic Reviews concluded that overall, there is little evidence that PUFA supplementation provides any benefit for the symptoms of ADHD in children and adolescents, although few data reported an improvement with combined *n*-3 and *n*-6 PUFA supplementation. However, the Authors underlined the urgency of future research prompted to correct the current weaknesses, such as small sample sizes, variability of selection criteria, variability of the type and dosage of supplementation, short follow-up times and other methodological weaknesses [208]. In particular, a possible factor possibly explaining trial failure could be represented by a too late period of intervention, so DHA supplementation at an earlier stage of development could result in better outcomes. Thus, a large body of work still needs to be done. Interestingly, it could also been hypothesized that the failure of these trials involving PUFA supplementation may also be related to an enhanced PUFA catabolism that could determine PUFA imbalances. In this regard, children with ADHD have been reported to have abnormal plasma fatty acid profiles [209] while PUFA intake was not different compared to healthy subjects [202,209,210,211]. A faster PUFA catabolism can be an explanation, since children with ADHD were shown to exhale higher levels of ethane, a metabolite of *n*-3 PUFA oxidation used as of non-invasive quantification of their metabolism [212].

From a neurochemical point of view, deficits in cortical dopamine neurotransmission have been reported in patients with ADHD. Alteration in dopamine neurotransmission has been associated with lower brain DHA concentrations [151]. It has been reported in rats that a diet deficient in *n*-3 PUFAs received for three weeks in utero and for two weeks post-natally caused profound changes in the dopaminergic system. In particular, it was shown a substantial enrichment in D2 receptors in discrete regions of the mesolimbic and mesocortical pathways as well as in a large number of brain areas of *n*-3 PUFA-deficient pups. The Authors hypothesized that such high expression of D1 and D2 receptors may be attributed to a behavioral hypersensitivity caused by the possible impairment of DA production during brain development [213].

Studies in animals fed with *n*-3 PUFA deficient diets identified many effects on dopaminergic systems, including lower levels of dopamine [152,214,215], of D2 receptors, of D2 receptor mRNA and dopaminergic presynaptic vesicles [152], and increased dopamine catabolism [216] in the PFC. Decreased tyrosine hydroxylase, the rate-limiting enzyme in dopamine synthesis, has been reported in offspring of rat dams fed with *n*-3 PUFA deficient diet soon after conception [217].

On the other hand, other indirect evidence for PUFA’s beneficial effect during gestation for ADHD comes from the study of Monk et al. [2] indicating that anxiety disorder in pregnant women is a risk factor for ADHD development in offspring. In this regard, low PUFA dietary intake has been linked to anxiety disorders [218], while *n*-3 PUFA supplementation showed anxiolytic effects [219]. Therefore, an adequate consumption of *n*-3 PUFAs during pregnancy may lead to beneficial effects in children also by reducing anxiety symptoms in childbearing women.

## 3. Conclusions

In the present review, we have summarized the main literature data regarding the potential role that maternal PUFA consumption may have on the risk for developing psychiatric complications in later adulthood. In this regard, maternal malnutrition and particularly *n*-3 PUFA deficiency have been linked to the development of neuropsychiatric conditions that become overt in early childhood, such as ASD; later childhood, such as ADHD; adolescence, such as SSD; or in adolescence and adulthood, such as depression and anxiety disorders. The influence of PUFAs in these conditions could be explained based on their significant effects on brain development and maturation.

*N*-3 supplementation during gestation and lactation periods has been shown beneficial in some studies, although many efforts need to be done in this regard. Better structured protocols and the choice of proper timeframe of interventions would surely lead to more consistent results.

In conclusion, the available results, although not completely conclusive, suggest that changes in lifestyle, particularly during pregnancy, could result helpful in preventing or reducing the risk of developing neuropsychiatric disorders later in life.

All these neuropsychiatric diseases share some common features, such as alteration in vulnerable brain region, especially in brain areas crucially involved in the etiopathogenesis of these pathologies, such as PFC or subcortical circuitries. Given the crucial role played of DHA in this area, especially during brain development and maturation, special attention should be paid in the choice of a correct diet. Many recommendations have been released by national and international authorities [111,220,221] and scientific research is frenetically working in supporting the crucial role that diet hold in prevention and possible treatments of neuropsychiatric diseases.

## Figures and Tables

**Figure 1 brainsci-06-00024-f001:**
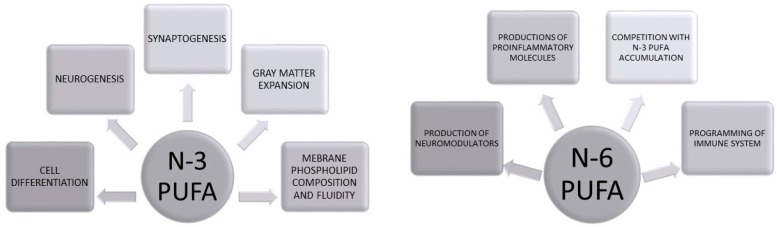
PUFAs in brain development and maturation. Figure 1 graphically depicts the main role of PUFAs in brain development (prenatal period) and maturation (postnatal period).

**Table 1 brainsci-06-00024-t001:** Main PUFA sources.

Type of PUFAs	Family	Food Source	References
α-linolenic acid	*n*-3	walnut, rapeseed, flax seeds and chia	[26]
docosahesaenoic acid	*n*-3	seafood	[27]
eicosapentaenoic acid	*n*-3	seafood	[27]
linoleic acid	*n*-6	soybean, canola, corn, safflower, sunflower and cottonseed	[22,25]
arachidonic acid	*n*-6	meat or products derived from animal fed with high *n*-6 content diets	[22,25]

PUFA: polyunsaturated fatty acids.

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
