# Peer review of "Maternal Malnutrition in the Etiopathogenesis of Psychiatric Diseases: Role of Polyunsaturated Fatty Acids"

_brainsci, 2016, doi:10.3390/brainsci6030024_

Round 1

Reviewer 1 Report

In this review Maria Grazia Morgese et al., have discussed the role of polyunsaturated fatty acids in the development of psychiatric disorders. The article is informative and mostly well written. However, there are few concerns that need to be addressed before being accepted for publication.

Reviewer comments

Oxidative stress is a major player in maternal stress and neurodevelopmental disabilities.  It would be interesting if the authors could add some discussion about if and how different PUFAs affect the oxidativee stress environment during neurodevelopment.

Lines 132-171 – It seems the focus of the review has been lost a bit in this section. Discussion on ELP could be focused on PUFA and its effects, by eliminating lines 132-171 or by shrinking it significantly.

The article lacks tables/figures. It could become more attractive if the authors could add a schematic representation (eg; PUFA targets in neurodevelopment) and one or two tables (eg; different food sources of PUFAs and differential effects of n-6vs n-3 fatty acids).

The conclusion section could expand to ‘summary and conclusion’. The authors could summarize important points from the review into a paragraph before concluding the review.

Please proof read lines 31, 33, 52, 54, 59, 67, 83, 84, 94, 196, 198, 209-210 for grammatical, spelling and typo errors

Lines 358-362 - Author contributions section needs to be rewritten. 

Line 64 – Please correct to “prefrontal cortex (PFC)

Please, consistently use ‘n6’, some places it is ‘n-6’.

Please correct “Authors” to “authors” in lines 285 and 315.

Author Response

We thank the Reviewer for the valuable suggestions.

Modifications to the previous version have been made according to Reviewer’s suggestions.

1.       Oxidative stress is a major player in maternal stress and neurodevelopmental disabilities.  It would be interesting if the authors could add some discussion about if and how different PUFAs affect the oxidative stress environment during neurodevelopment.

This section has been added accordingly, please refer to lines 99-120

2.       Lines 132-171 – It seems the focus of the review has been lost a bit in this section. Discussion on ELP could be focused on PUFA and its effects, by eliminating lines 132-171 or by shrinking it significantly.

This section has been reduced, accordingly.

3.       The article lacks tables/figures. It could become more attractive if the authors could add a schematic representation (eg; PUFA targets in neurodevelopment) and one or two tables (eg; different food sources of PUFAs and differential effects of n-6vs n-3 fatty acids).

We have added 1 table and 1 figure as indicated in the text, please see lines 51, 91-92 and 432-436.

4.       The conclusion section could expand to ‘summary and conclusion’. The authors could summarize important points from the review into a paragraph before concluding the review.

This section has been added accordingly, please refer to lines 405-415.

5.       Please proof read lines 31, 33, 52, 54, 59, 67, 83, 84, 94, 196, 198, 209-210 for grammatical, spelling and typo errors

Grammatical, spelling and typo errors have been corrected.

6.     Lines 358-362 - Author contributions section needs to be rewritten.

Author contributions section was rewritten, please see lines 428-429

7.     Line 64 – Please correct to “prefrontal cortex (PFC); 8. Please, consistently use ‘n6’, some places it is ‘n-6’; 9. Please correct “Authors” to “authors” in lines 285 and 315.

The corrections requested were made, accordingly.

Reviewer 2 Report

Re: Brainsci-134638

Morgese and Trabace have written a review of PUFAs in psychiatry.  The review is fairly comprehensive on the basic science side, but there are significant issues with the language.  I was trying to track the editorial issues for itemized feedback, but by the end of page 2, I already had over 30 grammatical and/or language problems listed, so I aborted this attempt.  Some of these errors are very significant, such as saying ‘rape’ instead of ‘rapeseed’, and any English reader would be struck by seeing the term for sexual assault being used due to language problems.  The authors ABSOLUTELY MUST work with an expert on the English language before this can be fully evaluated, as there are some places where the grammar/language issues make it challenging to read.

That said, the authors have contributed to the literature, and are appropriate to review this topic.  Regarding the content, I do have a few other comments:

1-      There should be a clear emphasis on the fact that in humans, the conversion to DHA is less efficient than in other animals such as rodents, which needs to be considered when trying to draw comparisons between rodent studies and human studies. There is some mention of this inefficiency in humans in lines 68-69, but this human-rodent contrast deserves specific attention as it has implications for the field.

2-      In the Early Life Programming section, there is discussion of maternal glucocorticoids in pregnancy, and prenatal stress is discussed for schizophrenia, but the authors have missed including the growing literature on prenatal stress in autism [J Autism Dev Disord, 2005, 35(4), 471-478; J Autism Devel Disord, 2008, 28, 481-488; Am J Epidemiology, 2005; 161, 916-925; Psychol Med  2014, 44(1), 71-84; Autism, 2016; 20(1), 26-36; and Hecht et al, Autism Research, Epub ahead of print PMID: 27091118].  Also missing is a related mouse model demonstrating autism-associated behaviors with a relative omega-3 deficiency during pregnancy [Jones et al. 2013, Behav Brain Res 238:193-199]  as well as the clinical studies suggesting relative omega-3 deficiency in autism [example Vancassel et al. 2001, Prostaglandins, Leukotrienes, and Essential Fatty Acids 65:1–7].

3-      The authors also do not adequately present the clinical trials literature for omega-3 in depression and anxiety, despite including the trials for ADHD.

4-      Late in the Depression and Anxiety section, the authors veer off into discussing degenerative diseases.  I think this is a perfectly appropriate discussion to include, but needs to be restructured to be under a separate heading.  Also – need to present an awareness that the ‘increase’ in Alzheimer’s in line 233 in eastern countries may be due to increased diagnosis, and not an actual change in incidence.

5-      Similarly, in the Schizophrenia section, the second paragraph was an interesting discussion of maternal DHA effects, but it doesn’t seem to belong in the Schizophrenia section.

6-      In the discussion of the reason for failed trials for ADHD, it seems that the most obvious issue was not discussed- that the DHA may need to be given earlier in development, especially as the discussion states ‘maternal low consumption of n3 PUFA or low dietary intake after birth may cause ADHD’ with cited references right before mentioning the failed trials.

7-      The first sentence in the Conclusion is overstated.  Please rephrase.  We can’t be so definitive in the recommendations at this point, although it is certainly important and more definitive studies need to be done, and MAY lead to this conclusion.

Author Response

We thank the Reviewer for the valuable suggestions.

Modifications to the previous version have been made according to Reviewer’s suggestions.

1.     Morgese and Trabace have written a review of PUFAs in psychiatry. The review is fairly comprehensive on the basic science side, but there are significant issues with the language. I was trying to track the editorial issues for itemized feedback, but by the end of page 2, I already had over 30 grammatical and/or language problems listed, so I aborted this attempt. Some of these errors are very significant, such as saying ‘rape’ instead of ‘rapeseed’, and any English reader would be struck by seeing the term for sexual assault being used due to language problems. The authors ABSOLUTELY MUST work with an expert on the English language before this can be fully evaluated, as there are some places where the grammar/language issues make it challenging to read.

The current version of the manuscript has been revised and proofread by an English speaker.

In regard to rape and rapeseed, we have modified in the text in order to do not confound the reader; anyway according to Encyclopaedia Britannica: “rape (Brassica napus, variety napus), also called rapeseed or colza”.

2.       There should be a clear emphasis on the fact that in humans, the conversion to DHA is less efficient than in other animals such as rodents, which needs to be considered when trying to draw comparisons between rodent studies and human studies. There is some mention of this inefficiency in humans in lines 68-69, but this human-rodent contrast deserves specific attention as it has implications for the field.

The issue has been further discussed, please refer to lines 73-76.

3.        In the Early Life Programming section, there is discussion of maternal glucocorticoids in pregnancy, and prenatal stress is discussed for schizophrenia, but the authors have missed including the growing literature on prenatal stress in autism [J Autism Dev Disord, 2005, 35(4), 471-478; J Autism Devel Disord, 2008, 28, 481-488; Am J Epidemiology, 2005; 161, 916-925; Psychol Med  2014, 44(1), 71-84; Autism, 2016; 20(1), 26-36; and Hecht et al, Autism Research, Epub ahead of print PMID: 27091118].  Also missing is a related mouse model demonstrating autism-associated behaviors with a relative omega-3 deficiency during pregnancy [Jones et al. 2013, Behav Brain Res 238:193-199]  as well as the clinical studies suggesting relative omega-3 deficiency in autism [example Vancassel et al. 2001, Prostaglandins, Leukotrienes, and Essential Fatty Acids 65:1–7].

The role of prenatal stress in Autism has been added, please refer to lines 329-333 and 340-341.

4.       The authors also do not adequately present the clinical trials literature for omega-3 in depression and anxiety, despite including the trials for ADHD.

The issue has been further discussed, please refer to lines 229-235.

5.         Late in the Depression and Anxiety section, the authors veer off into discussing degenerative diseases.  I think this is a perfectly appropriate discussion to include, but needs to be restructured to be under a separate heading.  Also – need to present an awareness that the ‘increase’ in Alzheimer’s in line 233 in eastern countries may be due to increased diagnosis, and not an actual change in incidence.

The issue has been further discussed, please refer to lines 279-287.

6.        Similarly, in the Schizophrenia section, the second paragraph was an interesting discussion of maternal DHA effects, but it doesn’t seem to belong in the Schizophrenia section.

We have better clarified the role of this point in the related section.

7.       In the discussion of the reason for failed trials for ADHD, it seems that the most obvious issue was not discussed- that the DHA may need to be given earlier in development, especially as the discussion states ‘maternal low consumption of n3 PUFA or low dietary intake after birth may cause ADHD’ with cited references right before mentioning the failed trials.

The point was added in the related section, please see lines 374-376.

8.       The first sentence in the Conclusion is overstated.  Please rephrase.  We can’t be so definitive in the recommendations at this point, although it is certainly important and more definitive studies need to be done, and MAY lead to this conclusion.

The sentence has been reformulated, please see lines 416-418.

Round 2

Reviewer 2 Report

Re: Brainsci-134638

Morgese and Trabace have written a review of PUFAs in psychiatry.  The manuscript is much improved regarding the specific items that had been missing in the discussion, as well as the language in the first 2 pages, but the use of this English editor seemed to decline after the first 2 pages.  Suggested edits:

1-      Line 145 ‘has dramatically increased during decades’- do the authors mean ‘has dramatically increased during the recent decades’?  In contrast to ‘during the decades of life’- which I don’t believe is intended.

2-      Line 167, change ‘source can derive’ to ‘sources can be derived’

3-      Line 172, change ‘hypothesis’ to ‘an hypothesis’

4-      Line 173, change ‘sensibly’ to ‘noticeably’

5-      Line 175, the new text, change ‘in respect to humans’ to ‘in contrast to humans’.

6-      Line 178, change ‘In human, during prenatal’ to ‘In humans, during the prenatal’.

7-      Line 179- not sure what is meant by ‘concomitance’ here.

8-      Line 190, change ‘5 folds’ to ‘5 fold’.

9-      Line 199- insert ‘the’ before ‘prenatal period’

10-   Line 233- what is meant by ‘in atypical location’?  That usually implies aberrant development, if it’s atypical, but it appears you are describing typical development.  What do you mean?  Also, grammatically should be ‘in an atypical location’ or ‘in atypical locations’.

11-   Line 238, change ‘the high’ to ‘a high’.

12-   Lines 253, 265, 278, 281, 287, 315, 345, 504 change ‘offsprings’ to ‘offspring’.

13-   Line 260, change ‘to an increased risk’ to ‘with an increased risk’. 

14-   Also lines 266, 272, 288, 294-295, 445 change ‘associated to’, to ‘associated with’.

15-   Lines 269-276 and the highlighted text in 280-285 are nearly identical in context.  Please avoid redundancy.

16-   Line 302, change ‘depend’ to ‘depends’

17-   Line 306, change ‘glucocorticoids’ to ‘glucocorticoid’.

18-   Line 313, change ‘inactive metabolite and this providing fetal protection toward’ to ‘an inactive metabolite and thus providing fetal protection from’

19-   Line 329, change ‘have’ to ‘has’.

20-   Line 369, and line 463 change ‘plasmatic’ to ‘plasma’.

21-   Line 391, remove ‘way’.

22-   Line 394, change ‘eziopathogenic’ to ‘etiopathogenic’.

23-   Line 421, change ‘reverting’ to ‘reversal’.

24-   Line 445, reference number is amiss ‘(23098794)’.

25-   Line 452, change ‘male’ to ‘males’ and ‘female’ to ‘females’.

26-   Line 461, change ‘is mainly occurring in children characterized’ to ‘mainly occurs in children and is characterized’.

27-   Line 478, change ‘justifying’ to ‘explaining’.

28-   Line 479, change ‘in’ to ‘at an’.

29-   Line 481, not sure you need ‘in this way’.

30-   Line 492, insert ‘the’ before ‘dopaminergic’.

31-   Line 502, change ‘PUFA’ to ‘PUFA’s’.

32-   Line 506, change ‘consuming’ to ‘consumption’.

33-   Line 518, change ‘way’ to ‘regard’.

34-   In general, several places in the mauscript- when using a plural, say ‘PUFAs’ instead of ‘PUFA’- like ‘PUFAs have’, not ‘PUFA have’.

Author Response

We thank the reviewer for the suggestions. We modified the manuscript accordingly.

In regard to redundancy in lines 269-285, lines 269-280 should have been already removed in the previous version, but for some reasons such changes were not implemented in the version  of reviewers. 

Hopefully, the current version is improved.